# Mechanistic multiscale modelling of energy metabolism in human astrocytes reveals the impact of morphology changes in Alzheimer's Disease

**Sofia Farina[1], Valérie Voorsluijs[2,3], Sonja Fixemer[2,4], David S. Bouvier[2,4,5], Susanne Claus[6], Mark H. Ellisman[7], Stéphane P. A. Bordas[1]\*, Alexander Skupin[2,3,7]\***

**1** Department of Engineering, University of Luxembourg, Esch-sur-Alzette, Luxembourg, **2** LCSB-Luxembourg Centre for Systems Biomedicine, University of Luxembourg, Esch-sur-Alzette, Luxembourg, **3** Department of Physics and Material Science, University of Luxembourg, Luxembourg, Luxembourg, **4** Luxembourg Center of Neuropathology (LCNP), Dudelange, Luxembourg, **5** Laboratoire national de santé (LNS), National Center of Pathology (NCP), Dudelange, Luxembourg, **6** Onera, Palaiseau, France, **7** Department of Neurosciences, University of California San Diego, California, United States of America

\* stephane.bordas@alum.northwestern.edu (SPAB); alexander.skupin@uni.lu (AS)

**Data Availability Statement:** The code and data associated with the simulations are available at the following repository: https://gitlab.lcsb.uni.lu/ICS-lcsb/energy-metabolism-model-astrocyte.

## Abstract

Astrocytes with their specialised morphology are essential for brain homeostasis as metabolic mediators between blood vessels and neurons. In neurodegenerative diseases such as Alzheimer's disease (AD), astrocytes adopt reactive profiles with molecular and morphological changes that could lead to the impairment of their metabolic support and impact disease progression. However, the underlying mechanisms of how the metabolic function of human astrocytes is impaired by their morphological changes in AD are still elusive. To address this challenge, we developed and applied a metabolic multiscale modelling approach integrating the dynamics of metabolic energy pathways and physiological astrocyte morphologies acquired in human AD and age-matched control brain samples. The results demonstrate that the complex cell shape and intracellular organisation of energetic pathways determine the metabolic profile and support capacity of astrocytes in health and AD conditions. Thus, our mechanistic approach indicates the importance of spatial orchestration in metabolism and allows for the identification of protective mechanisms against disease-associated metabolic impairments.

## Author summary

Among the cells in our brain astrocytes are crucial to support neurons. Understanding the role played by astrocytes in neurodegeneration is of the utmost importance. In particular, the relationship between morphology and spatial metabolic arrangements requires further investigation since astrocytes present morphological changes and metabolic impairments in neurodegenerative diseases. We propose a computational metabolic model for better understanding the interplay between these two aspects of astrocytes. The model is solved

**Funding:** S Farina and S Fixemer were supported by the PRIDE program of the Luxembourg National Research Found through the grants PRIDE17/12252781/DRIVEN and PRIDE17/12244779/PARK-QC, respectively. ME obtained support through NIH NINDS (1U24NS120055-01) and NIGMS (R24 GM137200). The funders had no role in study design, data collection and analysis, decision to publish, or preparation of the manuscript. No author received direct salary from any funder.

**Competing interests:** The authors have declared that no competing interests exist.

in domains of increasing complexity from a two-dimensional circular domain to three-dimensional human astrocytes. The findings emphasise the importance of the spatial arrangement of metabolic reaction sites for the metabolic dynamics, which is further emphasised in the intricate structures of astrocytes where their morphological changes can be crucial for facilitating metabolic dysfunctions in neurodegeneration.

## Introduction

The human brain is the organ with the highest energy demands required to sustain the high activity of neurons [1]. Astrocytes are multitasking glial cells directly contributing to brain homeostasis and metabolism. By their complex architecture as star-like branched cells, they are intermediate structures sitting between neurons and their synapses, which they enwrap with their intricate processes, and the blood vessels, which they contact with their endfeet. Based on this strategic positioning, astrocytes act as metabolic supporters providing energy in the form of lactate (LAC) to neurons and modulating their activity [2, 3]. While it is known that astrocytes provide neurons with lactate in high energetic demand, the hypothesis of astrocyte-neuron lactate shuttle (ANSL) [2] is an open debate [4, 5]. Astrocytes are also known to respond to brain "trauma" and drastically change in many brain diseases such as Alzheimer's disease (AD). In these situations, they engage reactive profiles with changes in morphology and in their molecular program [6, 7] like in AD where human astrocytes exhibit overbranching and context-dependent hypertrophy [8–11]. In addition to morphological changes, AD-associated astrocytes also exhibit metabolic dysfunctions [12–16], altering their role as neuronal supporters, but the relation to morphology is not established.

The metabolic support function of astrocytes depends on sufficient LAC production and efficient LAC export at the perisynapses as energy substrate for neurons [17] and on sufficient availability of adenosine triphosphate (ATP) for its own metabolic sustainability [18] requiring an ATP : ADP ratio at least larger than one [19]. Furthermore, physiological conditions for functional astrocytes are characterised by an approximate 10:1 ratio between LAC and pyruvate (PYR), the substrate for lactate production and mitochondrial activity, further indicating their metabolic support function [20]. Hence, astrocytes have to keep a balance between a LAC based "altruistic" support mode and a more "egocentric" self-sustainability characterised by a high ATP : ADP ratio. The mechanistic relationship between the observed disease-related modifications in morphology and metabolic dysfunctions is still to be characterised and whether morphology changes might represent a compensatory mechanism remains elusive.

Here, we develop a general interdisciplinary approach to systematically investigate the interplay between astrocytic morphology and energy metabolism in AD by a novel spatiotemporal *in silico* model that allows for physiologically realistic simulations by integrating complex morphologies obtained by high-resolution confocal microscopy and thereby addresses the impossibility of appropriate *in vivo* human astrocyte studies. Metabolic modelling has been extensively addressed in the literature at different levels *via* detailed genome-scaled metabolic network models [21] or *via* targeted dynamic models [22], including astrocytic metabolism [23–27]. All existing models neglect the spatial dimensions as they describe the metabolic processes through ordinary differential equations (ODEs). The underlying assumption that diffusion and reaction rates of metabolism are large enough to smear out spatial aspects are challenged by the complex morphology of astrocytes and an increasing amount of evidence for relocation of enzymes and other reaction sites in different conditions [28, 29]. To include spatial variations and geometric effects, we developed a metabolic model by means of a complex

reaction-diffusion system (RDS) in realistic three-dimensional (3D) morphologies obtained from high-resolution confocal microscopy images of astrocytes in *post mortem* brain samples of AD patients and age-matched control subjects [30]. The modelling framework incorporates two essential astrocytic properties: 1) the main reactions of glucose metabolism are spatially localised to reflect the heterogeneous distribution of enzymes in the cell [28], and 2) the complex and context-dependent geometry of cells is directly incorporated from high-resolution microscopy [10, 31]. To address the resulting computational challenges in solving the corresponding partial equations of the RDS in realistic astrocytic morphologies with thin branches and regions of high curvature and kinks, we adapted our previous approach [32] utilising the power of the cut finite element method (CᴜᴛFEM) [33, 34] to disentangle the complex astrocytic geometries from the mesh generation of finite-element methods and handle complex geometries as independently of the mesh as possible. Compared to our previously proposed model, we calibrated physiologically the parameters, investigated the spatial arrangements and molecular diffusivity and include real astrocytic morphologies.

By this approach, our model paves the way to more physiological modelling of the effect of astrocytic morphology in AD. Our framework is general and open-source and can be used for other cell types characterised by high-resolution imaging. For model establishment, we first performed simulations in simple two-dimensional (2D) geometries and studied how metabolic dynamics are affected by the spatial arrangement of reaction sites. The findings in 2D indicated the importance of the spatial component and the diffusion limitation that arises from the competition between the corresponding reaction centers for the metabolic substrates. Furthermore, the results highlighted the fundamental role of mitochondrial organisation for the metabolic output of the system. Based on these insights, we subsequently investigated spatiotemporal metabolic dynamics in real 3D human astrocytic morphologies by our multiscale modelling approach and demonstrate the potential of our framework to study metabolic dysfunction in AD-related reactive morphology of astrocytes.

## Results

To investigate the potential mechanistic link between morphology and energy metabolic activity, our model describes glucose metabolism by five main metabolic pathways (Fig 1A). Our proposed model prioritises a simple description of the chemical reactions to focus our study on the morphology and on the spatial component of the model. We describe glycolysis *via* two subsequent pathways where the first represents the ATP consuming and the second one the ATP producing reactions. The first pathway is catalysed by a set of enzymes (hexokinase, phosphoglucose isomerase, phosphofructose kinase and the fructose bisphosphate aldolase), which consume glucose (GLC) and ATP to produce ADP and glyceraldehyde 3-phosphate (GLY). In the following, we describe this pathway by a coarse-grained hexokinase (HXK) activity capturing the rate-limiting step of the glycolysis. In the second lumped reaction, these metabolites are transformed into ATP and pyruvate (PYR) by a second set of enzymes (glyceraldehyde phosphate dehydrogenase, phosphoglycerate kinase, phosphoglycerate mutase, enolase and pyruvate kinase), which we describe by the overall activity of the pyruvate kinase (PYRK), catalysing the irreversible step in the second part of the glycolysis. The generated PYR is subsequently metabolised into LAC by the lactate dehydrogenase (LDH) or used by mitochondrial metabolism to generate ATP. The mitochondrial metabolic activity of the Krebs cycle and oxidative phosphorylation is described by the coarse-grained effective reaction Mito. Finally, another effective reaction (act) accounts for various ATP-consuming processes associated with cellular activity. These metabolic pathways are put into a spatial context by distributing the corresponding reaction centers into a spatial domain.

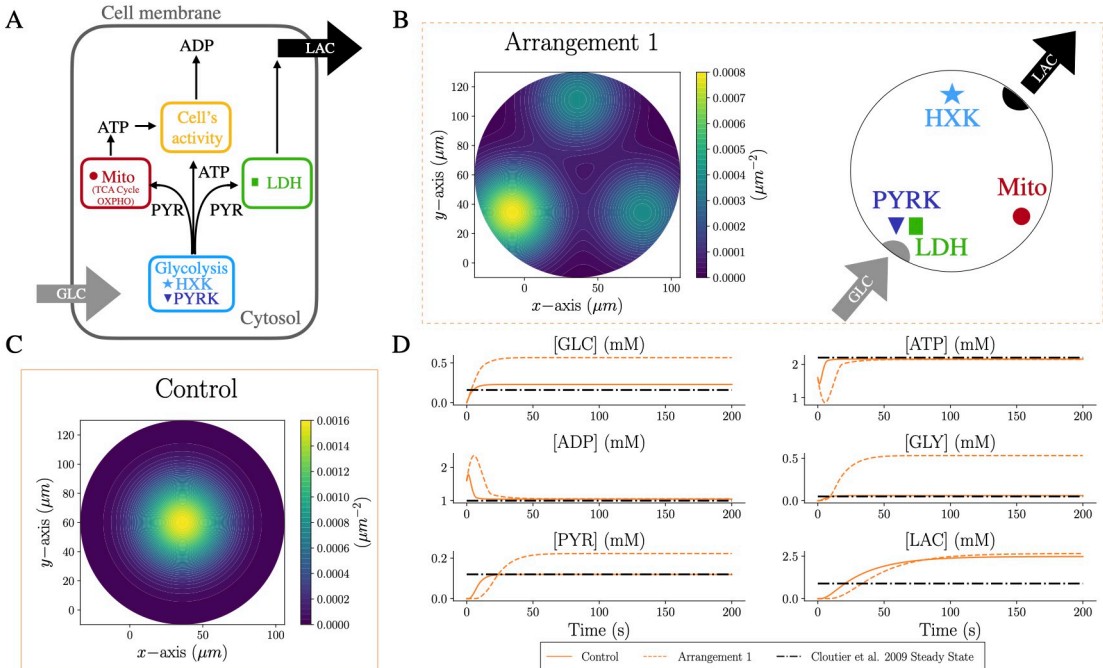

**Fig 1. Spatial arrangement of metabolism has an impact on cellular metabolite concentrations.** A: GLC enters the cytosol of the cell and takes part in glycolysis whose effective kinetics is captured by the two reactions HXK and PYRK. The products of glycolysis are subsequently consumed by the LDH reaction for generating LAC, by the act reaction describing ATP consumption due to cellular activity, and by mitochondria where the effective reaction Mito produces ATP from PYR through the Krebs cycle and oxidative phosphorylation. B: (left) Generic configuration to investigate the effect of metabolite transport on the output of metabolism in a 2D circular domain. The color map highlights the position of the reaction sites (defined by $\sum_i \mathcal{G}(\mathbf{x}_i, \sigma_i)$   $i = \{\text{HXK}, \text{PYRK}, \text{Mito}, \text{LDH}\}$), which are located on the vertices of an equilateral triangle. Two reaction sites are colocalised at the bottom left corner. (Right) Position of the reaction sites in Arrangement 1: HXK on the top close to the efflux of LAC, PYRK and LDH colocalised close to the GLC influx, and Mito on the last vertex. C: Control scenario: all reaction sites are located in the center. D: Dynamics of the average concentration of each species in Arrangement 1 and Control, compared with the steady-state values from Cloutier *et al*. [24].

## Metabolic dynamics and reaction sites competition in 2D domains

For model establishment and calibration, we first analyzed the effect of different spatial arrangements of reaction sites on the metabolic profile in simple 2D geometries. For this, we considered a circular domain and compared different configurations of reaction locations. The diameter of the circular domain was set to 140 $\mu$m as an average diameter that contains a full astrocyte [35]. To reflect the metabolic flux from the endfeet towards the perisynapses at the neurons' locations, we placed the entry of GLC and the exit of LAC at opposing sides of the circle (Fig 1B) where the subregions are defined as the intersection of a circle with a radius of 10 $\mu$m and centers are located at the origin for GLC and the antipodal point for LAC. The parameters used in the model are shown in Table 1 and sensitivity analysis can be found in S1 in S1 Text.

In this simplified setup, we first assumed that a given reaction occurs around a single location with a spatial extent of a Gaussian distribution with a width of $\sigma = 20.0$ $\mu$m except for the cellular activity. Biologically, cellular activity might also vary spatially [2, 36], with a higher energy demand at the perisynapses where neuronal activity induces higher cellular activity such as by neurotransmitter uptake. However, here we mainly focus on the metabolic reactions and their spatial relevance. As a control case ("Control"), all four reactions were located in the

**Table 1. Model parameters.**

| Model parameters | | | | |
|---|---|---|---|---|
| Parameter name | Value | Description | Units | Reference |
| $D_{\text{GLC}}$ | $0.6E3$ | diffusion coefficient of glucose | $[\mu m^2 s^{-1}]$ | [37] |
| $D_{\text{ATP}}$ | $0.15E3$ | diffusion coefficient of ATP | $[\mu m^2 s^{-1}]$ | [38] |
| $D_{\text{ADP}}$ | $0.15E3$ | diffusion coefficient of ADP | $[\mu m^2 s^{-1}]$ | [38] |
| $D_{\text{GLY}}$ | $0.51E3$ | diffusion coefficient of glyceraldehyde | $[\mu m^2 s^{-1}]$ | [39, 40] |
| $D_{\text{PYR}}$ | $0.64E3$ | diffusion coefficient of pyruvate | $[\mu m^2 s^{-1}]$ | [39, 40] |
| $D_{\text{LAC}}$ | $0.64E3$ | diffusion coefficient of lactate | $[\mu m^2 s^{-1}]$ | [39, 40] |
| $\text{GLC}(t=0)$ | $0.0$ | initial concentration of glucose | [mM] | |
| $\text{ATP}(t=0)$ | $1.6$ | initial concentration of ATP | [mM] | [24] |
| $\text{ADP}(t=0)$ | $1.6$ | initial concentration of ADP | [mM] | |
| $\text{GLY}(t=0)$ | $0.0$ | initial concentration of glyceraldehyde | [mM] | |
| $\text{PYR}(t=0)$ | $0.0$ | initial concentration of pyruvate | [mM] | |
| $\text{LAC}(t=0)$ | $0.0$ | initial concentration of lactate | [mM] | |
| $J_{\text{in}}$ | $0.048$ | influx of glucose | $[\text{mMs}^{-1}]$ | [24] |
| $J_{\text{out}}$ | $0.0969$ | degradation term of lactate | $[s^{-1}]$ | [24] |
| $K_{\text{HXK}}$ | $6.19E{-}02$ | reaction rate of hexokinase | $[(\text{mM})^{-2}s^{-1}]$ | [24] |
| $K_{\text{PYRK}}$ | $1.92$ | reaction rate of pyruvate kinase | $[(\text{mM})^{-2}s^{-1}]$ | [24] |
| $K_{\text{Mito}}$ | $8.13E{-}02$ | reaction rate of mitochondria activity | $[(\text{mM})^{-28}s^{-1}]$ | [24] |
| $K_{\text{act}}$ | $1.69E{-}01$ | reaction rate of cellular activity | $[s^{-1}]$ | [24] |
| $K_{\text{LDH}}$ | $7.19E{-}01$ | reaction rate of lactate dehydrogenase | $[s^{-1}]$ | [24] |

center of the circle as shown in Fig 1C, mimicking a well-stirred condition. In a more complex enzyme arrangement, we located the four reactions on the vertices of an equilateral triangle inscribed inside the circle: one reaction is placed on the top vertex close to the LAC exit, one reaction on the bottom right vertex and two reactions are placed on the bottom left vertex. An example is shown as "Arrangement 1" in Fig 1B, where PYRK and LDH are placed on top of each other, while HXK and Mito are on the top and bottom right vertex, respectively. The choice of this setting is arbitrary and not biologically relevant, with the only purpose of showing that the spatial arrangement of reaction sites impacts the output of the system.

The resulting dynamics of the Arrangement 1 and the Control setup are shown in Fig 1D where the average concentration dynamics inside the domain for each involved metabolite are shown. As a reference, we also plotted the steady state concentrations from Cloutier *et al.* [24], which was used to calibrate parameters of our model for which no literature information was available (see Table 1). As expected, the Control configuration leads to steady-state concentrations in agreement with Cloutier *et al.* during an equilibrium period of $\approx 50s$, with an exception for LAC which exhibits an almost doubled level. By contrast, the arrangement of the enzyme sites in spatially distributed configurations such as Arrangement 1 affects the metabolite levels of interest. For example, the steady-state corresponding to Arrangement 1 is characterised by concentrations of GLC, GLY, PYR and LAC that are approximately four, ten, two and three times higher compared to the well-mixed condition described by ODEs in Cloutier *et al.*, respectively.

The steady-state solutions of Arrangement 1 indicate the necessity of the species to diffuse inside the domain and reach the corresponding enzyme sites: GLC needs to diffuse into the other part of the domain to act as a substrate for HXK, and the produced GLY needs to reach the PYRK to be metabolised into PYR. The reactions are thereby diffusion-limited and the

system reaches the steady state before consuming more GLY. Finally, the increased LAC level for Arrangement 1 in relation to the Control is caused by the co-localisation of PYRK and LDH where produced PYR is directly metabolised into LAC whereas in the Control, Mito and PYRK compete for PYR as substrate.

To investigate systematically the effect of co-localisation and/or proximity of reaction centers to GLC influx or LAC efflux, we considered all possible location configurations for the four reactions on the vertices of the triangle (Fig 1B). Considering the colocalisation of two reactions in the left-bottom vertex leads to twelve possible location configurations (Fig 2A and 2C). As a first attempt to address slightly more complex morphologies, we studied the twelve arrangements within a two-dimensional star shape (Fig 2B) as a simplified version of an astrocyte. This setup allows for comparable results between the two domains since molecules have to pass similar distances between the subregions where GLC enters and the subregion where LAC is exported. Reaction sites were located analogously at the three vertices of an equilateral triangle within the star. As in the circular setup, we placed two reaction sites colocalised closer to the influx of GLC, and one reaction site at each of the remaining vertices.

Fig 2D shows the steady-state and spatially averaged concentration of each species of interest for the twelve possible configurations of the circular (left columns) and the star domain (right columns) where the maximum and minimum values for each species are highlighted in red and blue, respectively. Simulations performed in both domains exhibit similar trends. The species that are affected the most by the different spatial arrangement are GLC, GLY and PYR

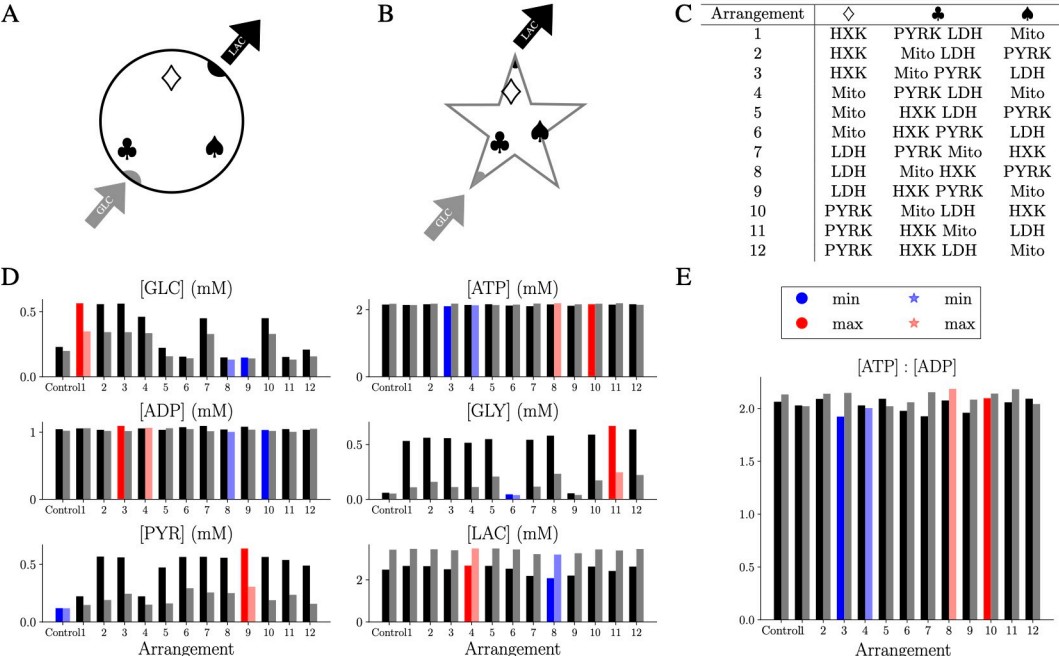

**Fig 2. Spatial organisation and competition between reaction sites affect the metabolic activity of the system.** A: Spatial setting of simulations performed in a 2D circle. GLC enters along one side, and on the diametrically opposite side, LAC is exported. Each of the three symbols is associated with one (diamond and spade) or two (club) reactions. B: The spatial setting of simulations performed in the star shape. The reaction sites are located analogously to the circle domain with the same distance between the GLC entry vertex and the LAC efflux/degradation. C: Table of the 12 possible configurations corresponding to the allocation of one reaction site to diamond and spade vertices, and two colocalised reaction sites at the club vertex. D: Spatially averaged steady-state concentrations of each species for the circle (left in black) and the star (right in grey) E: Spatially averaged steady state ATP : ADP ratio for simulations in a circular (left in black) and star-like geometry (right in grey).

for the co-localisation of the reaction sites HXK-PYRK (Location 6 and 9) and PYRK-LDH (Location 1 and 4) which led to low level of GLY or PYR, respectively. In the Control, where all the reaction sites overlap in the center of the domain, the system is more efficient with low levels of GLC, GLY and PYR, and a medium value of LAC. Although LAC shows differences depending on the location of the reaction sites, the changes are less significant due to the efflux which reduces LAC steady-state concentrations. Interestingly, the star-shape domain exhibits the highest values of LAC pointing to the importance of morphologies with branches and higher complexity. Since cellular activity is assumed to occur homogeneously inside the domains, variability in ATP and ADP levels across the setups are rather small confirmed by the [ATP] : [ADP] ratio with a variance between all the simulations of 0.005(mM$^2$) (Fig 2E).

Overall, this *in silico* experiments emphasise the variable output of the metabolic RDS as a function of the intracellular spatial organisation of reaction sites. To further investigate this effect, we next modelled the effect of enzyme distributions in more detail.

## Uniform and polarised distribution of reaction sites in a rectangular domain

Based on the establishment of the spatiotemporal metabolic model for one reaction center for each pathway reaction, we next explored the effect of inhomogeneous distributions of reaction centers on the metabolic state of the cell. For this purpose, we considered for each metabolic reaction ten distinct reaction sites with a smaller spatial extent ($\sigma = 1.0 \, \mu$m), while conserving the overall metabolically activity. To mimic the morphology of astrocytic branches, the shape of the RDS domain was chosen as a two-dimensional rectangle of dimension $[0, l] \times [0, L]$, with width $l = 4 \, \mu$m and a length $L = 140 \, \mu$m [35] where GLC enters from the bottom left corner of the rectangle (origin) and LAC exits from the top right corner. We considered two types of cellular organisation: one where the reaction sites are uniformly distributed inside the domain and the extreme opposite setting of a polarised cell where some reactions occur predominantly at one of the extremities of the cell. The polarised cells are a set of configurations representing an extreme enzymatic setting, which might not be physiologically relevant but allows us to isolate and test a mechanistic hypothesis. In other words, using extreme (and hence rather arbitrary) configurations is a way to validate a mechanism while avoiding the perturbations attributed to competing processes. To ensure the robustness of the findings, the two settings were compared by ensemble simulations of 200 distinct realisations of each setting. For the uniform cells, the coordinates of the 10 reaction sites of each type were randomly selected from a uniform distribution that covers the rectangular domain. realisations of polarised cells were generated either by normal distributions ($\mathcal{N}(m, \sigma')$, where $m$ and $\sigma'$ denote the mean and standard deviation, respectively) or by log-normal ($\log \mathcal{N}(m, \sigma')$) distributions. Fig 3A shows the position of the reaction sites along the $y$ coordinate of the 200 realisations and Fig 3B exemplifies enzyme distributions for a given cell for each setting. The different strategies for polarised cells lead to a certain probability for mitochondria localisation in the upper part of the domain for the "Polarised" configuration but not for the "Polarised $\log \mathcal{N}(2)$" configuration (Fig 3A). These settings allow for investigating the competition between the Mito and LDH reactions for their shared substrate PYR (S2 in S1 Text). In fact, the two kinds of polarised cells have been chosen to investigate the effect of different mitochondria locations. The glycolytic reactions are close to the influx to mimic the most efficient consumption of the glucose entering the cell, while the lactate dehydrogenase located on the opposite side of the domain allows the investigation of how the substrate pyruvate diffuses inside the domain and encounters first the Mito and then the LDH reactions.

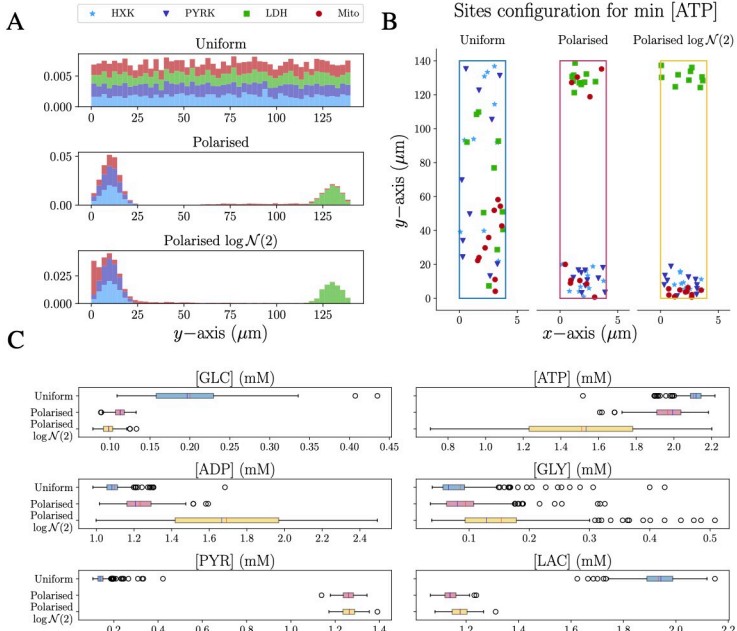

**Fig 3. The steady-state level of metabolites is affected by the polarised distribution of enzymes within cells.** A: Distribution of the reaction sites along the *y*-axis of the rectangular domain where the data are stacked on top of each other for visibility: HXK in light blue, PYRK in dark blue, LDH in green and Mito in dark red (top panel). In the Uniform setting, the reacting sites are uniformly distributed along the *y*-axis. In the Polarised settings, HXK, PYRK and LDH are spread unevenly over the domain with the first two located close to the origin and the latest close to the top of the domain. Mito reaction sites are distributed in the following way: 6 of them are normally distributed and colocated in the same area as HXK and PYRK, and 4 of them are uniformly located in the upper part of the domain (middle panel). In the Polarised log $\mathcal{N}(2)$ setting, mitochondria are located in the domain according to a log-normal distribution (bottom panel). B: Examples of Uniform, Polarised and Polarised log $\mathcal{N}(2)$ distributions for the less energised cell where mitochondrial production is the most affected by polarisation. C: Box plot of the average steady-state concentration of each species for the Uniform, Polarised and Polarised log $\mathcal{N}(2)$ distributions. (The mean and median of each box are signed in red and blue, respectively.).

To assess the effect of the different spatial arrangements, the steady state concentration of the 200 realisations, for the three different configurations, were compared statistically (Fig 3C) including T-test and Wilcoxon-Mann-Whitney with Holm-Bonferroni compensation (the corresponding table is shown in S3 in S1 Text). The p-value results show that only GLY for the uniform and polarised cell, and PYR for the two polarised configurations are not significantly impacted by the spatial arrangements where GLY exhibits a high *p*–value only in the t-test (0.093) but not in the non-parametric one. This finding is consistent with the similar average of the steady-state concentration but the distinct underlying distribution (Fig 3C). For PYR, the distributions for the two polarised cells exhibit a similar range (Fig 3C). However, since the two polarised cells only differ by the distribution of mitochondria it is expected that the distribution of PYR is similar because it is produced in PYRK reaction.

In general, the polarised cells consume more GLC than the uniform distributed ones, which is consistent with the fact that the reaction HXK is closer to the influx. GLY is present at a very low level for all configurations as also shown in the significance test. Interestingly, PYR and LAC differ strongly in polarised cells compared to the uniform setting with a higher level in PYR caused by faster metabolising of GLC by the HXK and subsequent PYRK reactions. On the other hand, LAC levels are higher for the uniform cells since in polarised cells PYR reaches the more distant LDH reaction only by the amount which has not been consumed by the closer

located Mito reactions. The resulting LAC : PYR concentration ratios for the cells with uniformly distributed enzymes respect the physiological constraints, whereas polarised cells exhibit ratios below one indicating an unphysiological or diseased state.

The corresponding ATP and ADP concentrations show a rather low variability for the uniform configuration with higher ATP and lower ADP concentrations (Fig 3C) compared to the polarised cells. Interestingly, the Polarised log $\mathcal{N}(2)$ configuration exhibits a very wide range for both concentrations with significantly different average values also in comparison with the Polarised configuration. Since the only significant difference between the Polarised and Polarised log $\mathcal{N}(2)$ configurations lies in the distribution of mitochondria inside the domain (red bars in middle and bottom panels of Fig 3A), we conclude that the difference in the cellular energetic state can only be attributed to the location of mitochondria within the cell. Furthermore, the ATP : ADP ratio for the three cellular configurations (Fig 4A) confirms that the Polarised log $\mathcal{N}(2)$ realisations cover an ATP : ADP ratio range from unhealthy (ratio < 1) to healthy (ratio $\geq$ 1).

The impact of the configurations on the metabolic activity and in particular with a focus on the "altruistic" behaviour producing more LAC or an "egocentric" strategy producing more ATP, can be visualised by the relationship between LAC and the ATP : ADP ratio (Fig 4B). We found two distinct clusters formed by uniform and polarised cells, where the uniform cells display a co-existing egoistic and altruistic mode characterised by high ATP and LAC concentrations for self-sustainability and neuronal support. Indeed, the correlation between the

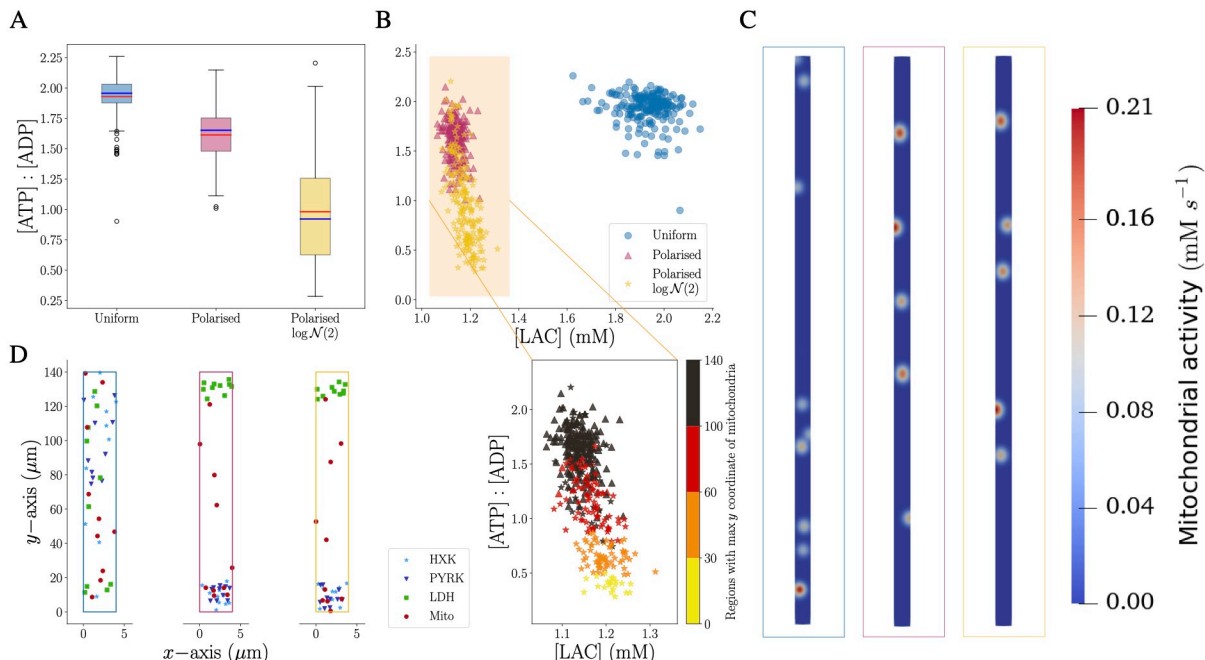

**Fig 4. Mitochondrial distribution determine [ATP] : [ADP] ratio and thereby energetic states of cells.** A: Box plot of the final average values of the [ATP] : [ADP] for Uniform, Polarised and Polarised log $\mathcal{N}(2)$ (Mean in red and median in blue). B: (top) scatter plot of the ratio against LAC final average values. There are two distinct clusters between the Polarised cells and the uniformly distributed ones. (bottom) Zoom on the ratio against LAC for Polarised cells colored based on the region where we can find the mitochondria with the highest $y$-coordinate. Interestingly if the enzymes are well distributed inside the domain, so if there is at least one mitochondria with a $y$-coordinate larger than 100 (black), the ratio value is higher. On the other hand, if the mitochondria are all located within the first or second region (yellow and orange), with $y$-coordinate lower than 60, the cell is in unhealthy status. C: Mitochondria activity of the configurations with maximum ATP production for the three types of cells at steady state where the activity indicates which mitochondria between the ones presented in panel D are producing ATP. D: Reaction sites setting for the maximum level of ATP for the three distributions.

variables, the ratio and LAC, is only slightly negative for the uniform cells (−0.19), whereas the group of polarised cells exhibits a stronger negative correlation (−0.65) indicating that high values of one quantity lead to low values of the other. "Polarised" cells are located on the top of the cluster and the "Polarised $\log \mathcal{N}(2)$" cells are predominantly in the lower part of the cluster characterised by a lower ATP : ADP-ratio and slightly higher LAC concentrations (Fig 4B). This difference indicates the importance of mitochondria localisation as shown by the color-indicated classification of the vertical arrangement of mitochondria. We colored the metabolic profile of each realisation based on the highest $y$–coordinate of Mito sites ($y_{max}$): yellow, orange, red or black, if $y_{max} < 30$, $30 < y_{max} < 60$, $60 < y_{max} < 100$ or $y_{max} > 100$, respectively. This analysis highlights that the lowest level of ATP coincides with realisations where all mitochondria are grouped in the lower region. By contrast, simulations with a high energetic profile correspond to arrangements where mitochondria are distributed throughout the whole rectangular shape. Fig 4C and 4D show the mitochondrial spatial arrangement for the most energised cells and the corresponding mitochondria activity at the steady state. Not all mitochondria shown in the spatial arrangements are active at the steady state. Their activity relies on the availability of ADP to produce ATP. For this reason, the mitochondria located in the lower part of the Polarised cells are not active (more in S4 in S1 Text). Hence, the mitochondria activity for these arrangements remarks the necessity of mitochondria to be well-distributed in the whole domain to sustain high ATP levels. The importance of having Mito reactions uniformly distributed inside the domain lies in the fact that Mito requires ADP as a substrate to be activated. However, the non-uniform distribution of the reaction sites, for example, the ones of the less energised cell with minimum ATP shown in Fig 3B, generates regions with a high gradient of ADP or ATP (more in S4 in S1 Text).

Overall, this analysis demonstrates the impact of the interplay between spatial enzyme orchestration and morphology on the metabolic profile of cells. Our finding highlights that different cellular organisation leads to different steady-state concentrations which might be linked to potential disease of cells.

## Morphological effects on the metabolic activity of human astrocytes in health and AD

Finally, we extend our work to 3D reconstructions of human astrocytes acquired from GFAP-immunostained *post-mortem* brain samples from age-matched control subjects (Fig 5A–5C) and AD patients (Fig 5D–5F). The 3D confocal images of the astrocytes were acquired in the CA1 subregion of the hippocampus (Fig 5A and 5D). Given the typical *post-mortem* nature of such brain samples, the dynamical consequences of the morphology for metabolic profiles can be only assessed by an appropriate *in silico* strategy. The respective segmentations of the prototypical astrocytes (Fig 5B and 5E) reveal significant differences in the volume and morphological diversity of the two cells: the reactive AD astrocyte exhibits hypertrophy [10], proliferation of branches and coverage of wider spatial domains in comparison with the less complex shape of the control astrocyte (Fig 5G and 5H). Based on mitochondria staining and segmentation (Fig 5A and 5B and Fig 5D and 5E), a realistic spatial arrangement of mitochondria is implemented in the multiscale model (Fig 5C and 5F). The presence of regions with different mitochondrial densities is respected by tuning the center positions and variances of the Mito spatial reaction rates (Fig 5C and 5F). The minimum variance is set to 1.0 $\mu$m and we scale accordingly the size of the other regions with a maximum of 2.0 $\mu$m. The number of reaction sites for the other reactions is set according to the amount of mitochondria selected from post-processing, 97 for the control and 140 for the reactive astrocyte (Fig 5I). For each Mito reaction site, we located a HXK site close by in agreement with the observed relationships between these

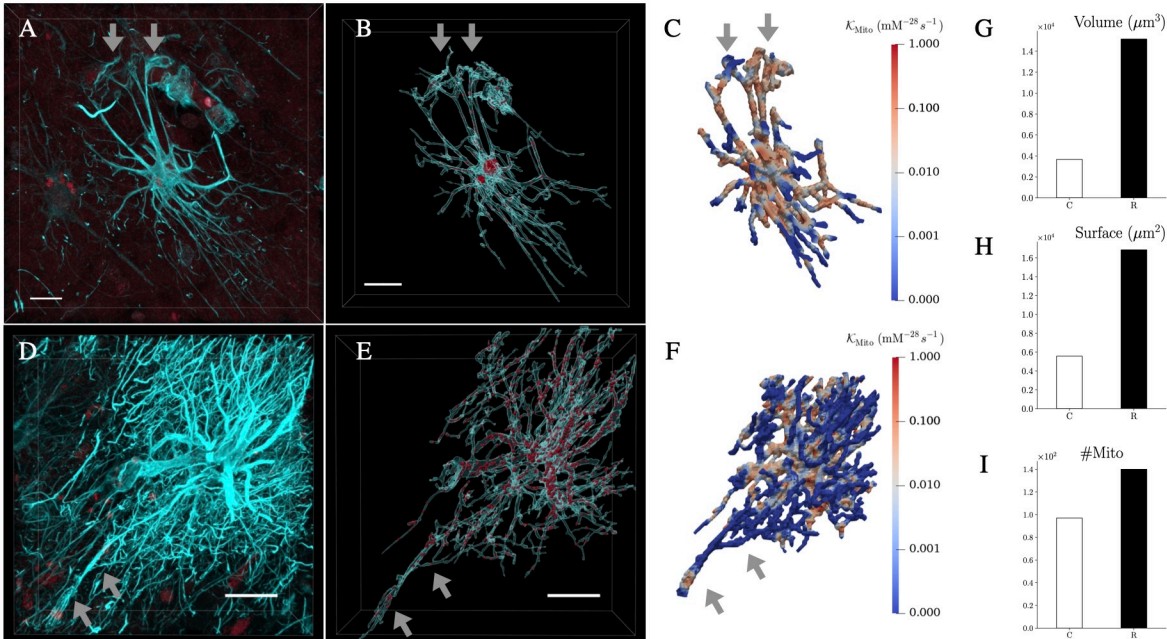

**Fig 5. Human hippocampal astrocytes from an age-matched control subject and AD patient: From microscopy image to 3D simulation setting.** A and D: High-resolution confocal microscopy images from an age-matched control subject (panel A) and from an AD patient (panel D) were obtained from 50–100 $\mu$m brain sections that were immunostained against GFAP (cyan) to visualise astrocyte cytoskeletal morphology, and against TUFM (dark red) to reveal mitochondria in the hippocampus where the grey arrows indicate the endfeet of the two astrocytes in accordance with the blood vessels locations. B and E: Using Imaris 9.6.0, astrocyte 3D morphology was segmented using the surface tool and mitochondria were labelled with the spots tool for the astrocytes in both conditions. Finally, based on the segmentation, we created the domains for our simulations and we selected the locations with higher density to define the Mito reactions. C and F: Spatial reaction rates $\mathcal{K}_{\text{Mito}}$ describing the mitochondria activity inside the cells. In the bar charts, we compared G: the cell volumes—3673 $\mu$m$^3$ for the control astrocyte (C) and 15161 $\mu$m$^3$ for the reactive (R), H: cell surfaces—5569 $\mu$m$^2$ for (C) and 16854 $\mu$m$^2$ for (R), and I: the number of mitochondria activity centers—97 for (C) and 140 for (R) computed in panel C and F. Scalebars: A-B 15 $\mu$m, D-E 30 $\mu$m.

two enzymes [41, 42]. The reaction sites of PYRK and LDH are taken from a uniform distribution defined in the three-dimensional box containing the astrocyte. The locations of the reaction sites for the simulations inside the control and AD reactive astrocyte are shown in Fig 6A and 6C together with the assumed endfeet for GLC influx and the subregions at the perisynapses for LAC export into the extracellular space (Fig 6A–6C). Since astrocytes are in contact with thousands of neurons [43] providing them with lactate, it would be biologically more accurate to consider multiple regions where lactate can exit the astrocyte [44]. In S5 in S1 Text, we provide additional experiments where we show the effect of considering more lactate exit sites and an increased cellular activity in the perisynaptic regions [2, 36]. However, in this section, we remain consistent with the previous experiments considering only a polarisation of the cell with lactate export regions on the opposite side of the glucose influx and a homogeneous cellular activity.

As a first analysis, we ran three baseline simulations based on the physiological parameters (Table 1) with one simulation inside the protoplasmic control morphology (C) (Fig 6A), one within the same morphology but with a polarised distribution of reaction centres (P) (Fig 6B) and one inside the reactive astrocyte (R) (Fig 6C, more details are given in S6 in S1 Text). The resulting dynamics of these baseline simulations (Fig 6D) are in good agreement with the investigation of enzyme distributions in the 2D domains where scenarios C and R resemble

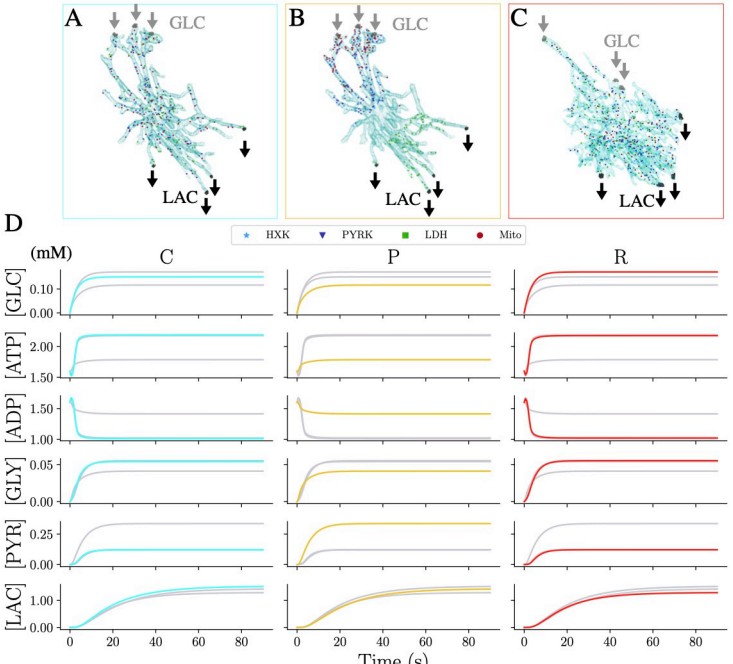

**Fig 6. Metabolite dynamics in 3D astrocytes with physiological reaction site versus extreme polarised arrangements.** Setting of the 3D simulations for the A: control (C), B: polarised (P) and C: reactive (R) astrocyte. For C and R, Mito reaction centers were inferred from the microscopic images. Each HXK site is sorted from a gaussian distribution centered at each Mito site. In this way, for each Mito we have an HXK reaction close by. PYRK and LDH are uniformly distributed inside the box that contains the cells. The reaction centers of P are sorted in the way that HXK and PYRK are colocalised close to the GLC influx, while on the other extremity of the cell, we locate LDH centers. Mito centers are sorted using a log-normal distribution that locates them in the same region as HXK. The number of centers per reaction type is 90 for C and P, and 140 for R. For the three settings, GLC enters three sub-regions from the branches of the cell in contact with the blood vessels and LAC exits from four sub-regions at the other extremity of the cell. D: Time behavior of the average concentration of each species for C (cyan), P (yellow) and R (red).

properties of the uniform distributed cell and P corresponds to the polarised cell (Fig 3B). However, the average LAC concentration is higher than expected for P and lower than expected for C and R. Also the PYR concentration is closer to that of the uniform setting for the P configuration. While the concentration values of C and R are on average very close (Fig 6B), smaller differences are visible mostly in GLC and LAC and attributed to the effect of the morphological differences and reaction site configurations.

To investigate a reactive astrocyte subject to AD, we extended our simulations by gradually adding AD-related dysfunctions. Experiment 1 (E1) mimics a loss of GLC uptake [45] by a 30% decreased GLC influx. Experiment 2 (E2) includes the dysfunction in mitochondrial activity [46–48] inducing a lower ATP production by a reduced reaction rate for Mito ($K_{\text{Mito}}10^{-5}$). In accordance with available experimental data [12], we considered an increment of the activity of LDH by a factor of ten in Experiment 3 (E3) and an increment in the glycolysis rate in particular in the PYRK reaction, also by a factor of ten in Experiment 4 (E4). In the final experiment (EAD), all four conditions were combined to explore their possible synergistic effects.

Fig 7A exhibits the percentage of the concentration loss at steady state for experiments E1, E2, E3, E4 and EAD compared to R. Interestingly, the 30% reduction in GLC uptake in E1 is

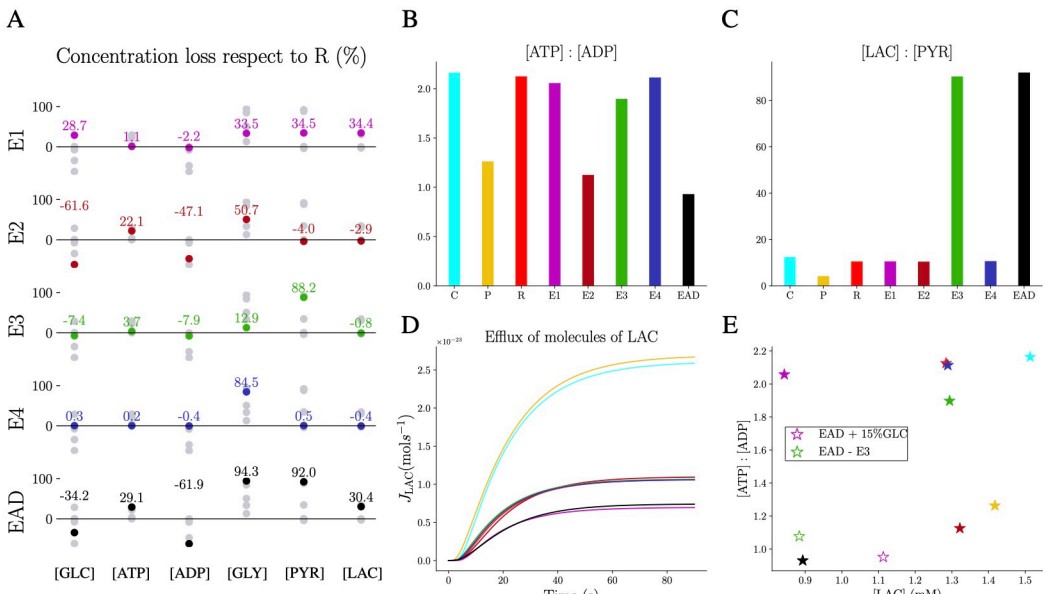

**Fig 7. Metabolite dynamics in 3D astrocytes with physiological reaction site versus extreme polarised arrangements.**
We consider four pathological conditions of AD, in the setting of the reactive astrocyte R. E1 describes the deficiency of
GLC uptake (magenta); E2, the mitochondria dysfunction (dark red); E3, the LDH overwork (green); E4, PYRK overwork
(blue) and EAD, the four conditions combined (black). A: Final average concentration loss with respect to R. The
experiments reflect their loss/gain imposed on the cell through the conditions. Steady-state spatially averaged B: ATP : ADP
ratio and C: LAC : PYR ratio of control (cyan), polarised (yellow), reactive (red) and all the AD experiments. D: Efflux of
LAC molecules exported over time from the astrocyte to the extracellular space. The experiments with higher export are the
two control astrocyte with C and P configurations. The experiments with a lower export are E1 with a loss in GLC uptake
and EAD with the combination of the AD conditions. E: Scatter plot of ATP : ADP against LAC final average values. The
most efficient cell is the control one. Then, the different AD conditions affect the cell status leading the reactive cell affected
by all the AD conditions to an unhealthy state. In order to save the EAD cell, we increase the uptake of GLC up to 85%
(white star with magenta edge), and the cell responds by using the more available fuel to produce more LAC. However,
blocking the LDH overwork (white star with green edge) increases ATP : ADP and thereby rescues the astrocyte from the
AD conditions.

reflected by the final steady state in GLC ($\approx$ 28.7% loss) which induced a loss of $\approx$ 35% in
GLY, PYR and LAC. Dysfunctional Mito reactions lead to an increase in the final GLC level
and a loss in ATP and GLY whereas the level of LAC is not affected. The experimentally
observed increased activity of LDH (considered in experiment E3) results mainly in faster
metabolising of PYR. On the other hand, GLY consumption is maximised by the turnover of
PYRK in the E4 experiment while the other concentrations are not affected. The combined
effect of the individual dysfunctions in the EAD experiment leads to a significant change in the
metabolic profile (with the highest loss in ATP, GLY and PYR Fig 7A). (The dynamics of these
experiments are shown in S7 in S1 Text.).

The functional state of cells in terms of ATP : ADP and LAC : PYR ratios at steady state is
preserved for a wide range of conditions. Even for the polarised P configuration, the ATP :
ADP ratio is higher than 1.0 (Fig 7B and 7C), suggesting that a complex shape makes the cell
more robust against extreme situations. This is also confirmed by the ratios of the E2 experi-
ment, which does not exhibit a ratio below 1.0 despite mitochondrial dysfunction. The only
cell that reaches a critical unhealthy state is the EAD condition (0.93), where mitochondrial
dysfunction adds to the other dysfunctions. Also, the ratio of LAC : PYR is always within the
physiological range ($>$ 10) for all conditions except P. However, a LAC : PYR ratio of above

80 is reported in E3 and EAD, which may indicate hypoxia with low levels of oxygen in blood [20].

Since LAC export into the extracellular space is an essential mechanism of astrocytic support to neurons, we also quantified LAC efflux exporting LAC from the corresponding subregions (Fig 7D). The asymptotic behaviour of the efflux indicates that cells with the C and P configuration export more LAC, suggesting that the less ramified morphology of the protoplasmic astrocyte allows for faster diffusion of molecules and subsequent export regions. On the other hand, E1 and EAD configurations export less, indicating that the 70% decrease in GLC uptake might drive this AD symptom. The different metabolic states of the cell are also assembled in the "altruistic" vs "egocentric" map in terms of the LAC concentration and the ATP : ADP ratio (Fig 7E). This map indicates the C configuration as the most efficient cell with high levels for both in agreement with the previous finding on uniform distributed cells. The P setup exhibits a more altruistic behaviour than expected by producing more LAC than ATP, potentially facilitated by the morphology. When cells lack GLC, they become more egoistic and produce more ATP. Remarkably, the steady state of LAC of the R, E2, E3 and E4 experiments is $\approx 1.3\,\mu$M but the ATP concentration is decreasing from high levels in the R and E4 configuration to lower concentration in E2. Finally, lower levels of both ATP and LAC are the AD-related EAD condition suggesting that it can neither support neurons nor itself. Last, we studied how to support an AD-impacted astrocyte where the results of the individual conditions helped to disentangle the different effects. Importing more GLC (by increasing the uptake to 85% of the healthy control condition) turns the cell into a more altruistic state by using the additional fuel predominantly for LAC production. Blocking the excessive activity of LDH saves the cell from AD-related energy deprivation but with the cost of reduced LAC export.

To investigate the impact of diffusion limitation as an underlying mechanism in reactive astrocytes, Fig 8 illustrates the time evolution of the 3D distribution of concentrations for the healthy C and AD-related EAD condition considering the properties and spatial distribution of reaction sites (S1–S12 Movies). In particular, the trapping effect discussed above is highlighted in the reactive astrocytic morphology for ATP and PYR where branches exhibit a higher concentration variability.

To summarise, the physiologically realistic simulations reproduce important features of astrocytes in healthy and diseased conditions. The incorporation of real morphologies highlights cellular robustness against extreme enzymatic configurations. This is also seen for AD conditions, indicating the influence of the cellular domain on the metabolic state of the cell. In fact, a single AD characteristic does not lead to an unhealthy cell, only combinations of AD terrain lead to severe metabolic dysfunctions.

## Discussion

Although the link between cellular morphology and metabolic activity might have implications for neurodegeneration including Parkinson's disease and Alzheimer's disease, our understanding of this connection remains imperfect due in part to experimental limitations. Moreover, the role played by astrocytic metabolism in neuronal support is an open discussion [49]. To address this challenge, we developed a multiscale model for energy metabolism in complex cellular domains with a specific focus on the intracellular spatial orchestration of astrocytes. To build the mathematical model, we first considered a single reaction site for each metabolic subpathway in a 2D circular geometry and validated the model in terms of physiological concentration ranges for astrocytes (Fig 1), in accordance with previous ODEs model [24]. We showed numerically that different spatial organisations of reaction sites lead to distinct

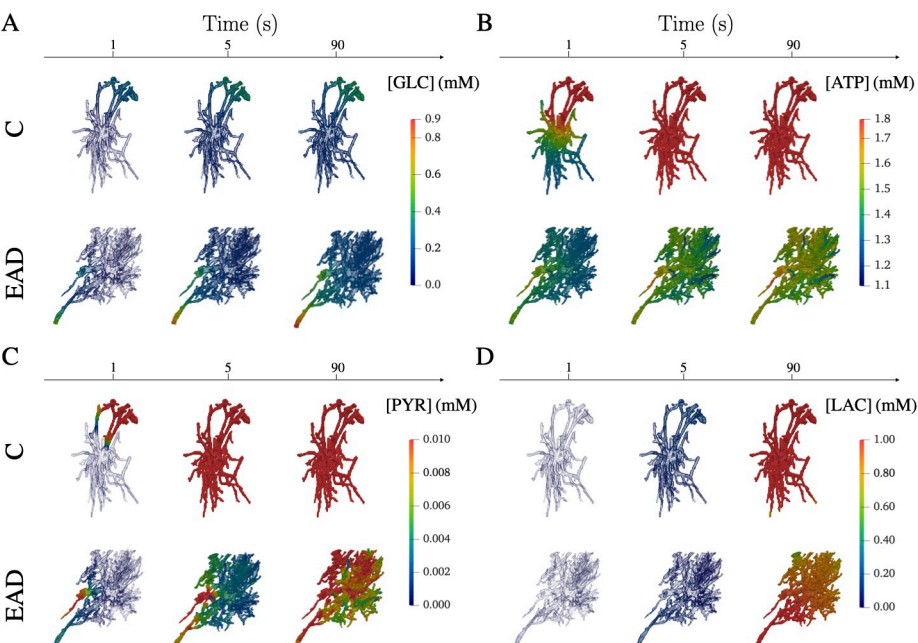

**Fig 8. Spatially resolved Control and EAD astrocytes for GLC, ATP, PYR and LAC at different times.** 3D spatial concentration of metabolites at three different time steps in control (C) and reactive astrocytes affected by AD pathology (EAD). A: GLC enters from the blood vessels and spreads inside the astrocytic domains activating glucose metabolism. B: ATP, already present in the cells at the initial time, is produced and consumed. In particular, in correspondence with regions with high numbers/absence of Mito sites, we can notice high/low levels of ATP in EAD. C: PYR produced by PYRK diffuses inside the 3D domains and highlights the complex shape of the reactive astrocyte with high variability of concentration within the cell. D: LAC shows a slow production, in fact at time 5, both C and EAD show low concentrations. At the final time, we can appreciate the steady state level of LAC where the regions where it is exported are highlighted by lower concentrations.

metabolic profiles due to diffusion limitation and local substrate competition (Fig 2). The observed differences between the circular and the star-shaped domain indicated a possible trapping effect for molecules in more complex shapes. These trapping effects might be overestimated compared to a more physiologically realistic astrocyte since many more reaction sites are typically present within an astrocytic branch. Nevertheless, these results strongly indicate that the spatial dimension and domain complexity can have a crucial effect on metabolic profiles and may be of particular importance for the metabolic support function of astrocytes.

To further characterise these spatial effects in a more physiological setting, we considered a larger number of reaction sites, which were distributed either within a uniform or polarised arrangement inside a rectangular shape, mimicking an astrocytic branch. For each configuration, we ran 200 realisations, allowing for robust statistical comparisons between the different settings (Fig 3). The results showed that cells with uniformly distributed reaction sites are significantly more efficient in both the "altruistic" LAC production as well as the "egocentric" intracellular energy state. Although polarised organisation corresponds to an extreme and rare biological setting, the analysis of these realisations indicates the importance of a more homogeneous mitochondria distribution for sufficient activity and a related energised cell state (Fig 4). This is in line with the experimental observation of mitochondrial organisation and homeostasis including fission and fusion where impairment of these processes is linked to neurodegeneration [50].

Based on the 2D model, we extended our investigations to physiological 3D morphologies of astrocytes, obtained from confocal microscopy images of human *post-mortem* brain samples

of an AD patient and a healthy control subject. Our approach is thereby able to integrate directly the spatial orchestration of reaction enzymes as demonstrated by the experimentally quantified mitochondrial distribution (Fig 5). We first confirmed that using different morphologies but the same parameters leads to concentrations in the physiological range in agreement with the findings in the simplified 2D geometries (Fig 6). To investigate the effect of AD-related molecular modifications, we analysed a reactive astrocyte with baseline parameters and four individual metabolic dysfunctions linked to AD and their combinations (Fig 7). The results highlighted that different pathological effects arouse specific system responses and differentiated the cell behaviour between an "altruistic" and an "egocentric" mode. Furthermore, the results indicated that any given dysfunction does not lead necessarily to a dysfunctional cell with a low ATP : ADP ratio but it is the cumulative metabolic insufficiencies that lead the cell into a critical state. This synergistic phenotype might be related to the multi-hit perspective in neurodegeneration which addresses the transient compensation and typical disease onset at higher age [51, 52]. The systematic study of the individual dysfunctions allowed to suggest that reducing LDH activity could sustain astrocytic function. Such approaches are also discussed in the context of cancer [53, 54]. However, in the context of AD, the challenge would be to interfere with metabolism in a cell-type-specific manner.

The comparison between the simplified 2D domains and the complex 3D morphologies indicates that real astrocytic shapes affect the cell state with robustness towards enzyme orchestration and different metabolic dysfunctions. This robustness might be caused by the trapping of molecules in thin branches as further indicated by the analysis of 2D star-shaped morphology. The thin processes may hamper the diffusion of molecules as shown by the spatial concentration profiles (Fig 8) which increased mitochondrial activity and corresponding ATP production with the cost of decreased LAC export. Thus, the complex morphology might provide a mechanism to support an "egocentric" state if the system reaches limiting conditions, similar to energy buffering in complex mitochondrial morphologies [55].

While in literature several modelling approaches for metabolism have been proposed, such as genome-scale modelling [21, 23, 56, 57] or more quantitative description via kinetic models [58–60], we propose a metabolic model including molecular diffusivity and spatial orchestraton of reaction sites. Furthermore, to our knowledge, our approach is the first 3D model of cellular energy metabolism using physiological human cellular morphologies. However, the crucial role of geometries has been taken into account in other modelling such as for calcium signalling in astrocyte [61–63] further indicating the importance of morphology. In agreement with these findings, our analyses of hippocampal control and AD-related reactive astrocytes clearly demonstrate the importance of morphology for cellular metabolic activity. While our studies refer to two specific astrocytic morphologies, an AD and an aged-match control, our framework can be easily applied to each segmented astrocytic image and the crucial role of complex morphology is already visible in these experiments. Our approach has limitations, such as the lack of cellular compartmentalisation, the coarse-graining of enzymatic reaction into effective metabolic pathways, the limitation of the GFAP staining and the incomplete information on reaction site localisation provided by imaging modalities.

The model can be expanded by considering more detailed equations and including pyruvate exchanger with mitochondria, creatine phosphate shuttle or oxygen concentration. The more complex mathematical formulation of the model can integrate the inhibition of glycolysis in the presence of a high concentration of ATP [64], which would allow us to investigate better the energetic behaviour of astrocytes. Also, we could extend the experiment presented in S5 in S1 Text, considering the cellular activity consuming more ATP in the perisynaptic regions [2, 36]. Following the experiment presented in S5 in S1 Text, the regions exporting lactate can be defined better for example by investigating the distribution of lactate transporters

via immunohistochemistry (IHC) images [65]. Furthermore, GFAP staining does not capture the full astrocytic morphology [31]. Comparison between GFAP staining and volumetric analyses by electron microscopic images (EM) have shown that volumes can differ by a factor of up to 1.5 which will particularly affect diffusion-mediated processes. This limitation can be addressed by appropriate scaling of the morphological reconstruction or adaptation of the diffusion coefficients. While such scaling may lead to minor quantitative changes in our analysis the qualitative findings will not be affected. This effect will be addressed in future investigations using 3D electron microscopic images which reconstruct astrocytic morphologies in more detail [66]. However, the images obtained by GFAP staining are suitable for creating finite element meshes, while EM images will require a more computationally demanding mesh construction. Concerning the computational model, a further step would be to ensure the accuracy of the solution by measuring the discretization error using real-time error estimation for biomedicine [67–70] or expanding the sensitivity analysis presented in S1 in S1 Text with detailed quantification analysis [71, 72].

Another important application of our approach will be the investigation of mitochondrial turnover and resulting distributions in astrocytes [73, 74]. Despite these limitations, we demonstrate the general importance and feasibility of physiological simulations by integrating molecular properties, spatial intracellular orchestration and morphology. Even though we characterised our results on the average steady-state concentration, further investigation can be done in subdomains as we showed for LAC in Fig 7D. Based on our multiscale framework, future investigations will allow disentangling different mechanisms underlying neurodegeneration, including mitochondrial morphology [55, 75], organisation and dysfunction [76–78] by more detailed models.

## Materials and methods

### Ethics statement

Post-mortem brain tissue was obtained from the Douglas-Bell Canada Brain Bank and handled according to the agreements with the Ethics Board of the Douglas-Bell Brain Bank (Douglas Mental Health University Institute, Montréal, QC, Canada) and the Ethic Panel of the University of Luxembourg (ERP 16–037 and 21–009). The two hippocampal samples used in this work were donated from a male 87-year-old Alzheimer's Disease patient with a disease stage of A2B3C2 and a post-mortem interval of 21,75 hours, and by a female 89-year-old (age-matched) control subject with a post-mortem interval of 23,58 hours. All donors gave their written consent.

### Energy metabolism model

The core energy metabolism is broken down into the core metabolic pathways by the coarse-grained non-reversible reactions:

$$\text{HXK} := \text{GLC} + 2\,\text{ATP} \rightarrow 2\,\text{ADP} + 2\text{GLY} \tag{1}$$

$$\text{PYRK} := \text{GLY} + 2\,\text{ADP} \rightarrow 2\,\text{ATP} + \text{PYR} \tag{2}$$

$$\text{LDH} := \text{PYR} \rightarrow \text{LAC} \tag{3}$$

$$\text{Mito} := \text{PYR} + 28\,\text{ADP} \rightarrow 28\,\text{ATP} \tag{4}$$

$$\text{act} := \text{ATP} \rightarrow \text{ADP}\,, \tag{5}$$

where the first two reactions consider the ATP-consuming and ATP-producing parts of gly-colysis, LDH describes the activity of lactate dehydrogenase. Mito reflects the overall metabolic activity of mitochondria in terms of ATP production and general cellular activity is reflected by the act reaction.

## Reaction diffusion system

To investigate the spatial coupling of the metabolic pathways (Eqs (1)–(5)), the reactions were integrated by a RDS [79]. The domain of the PDEs is a bounded subset of $\mathbb{R}^d$ ($d = 2$ or $3$), denoted by $\Omega$ and concentrations $[\cdot]$ are defined as function $[\,\cdot\,] : \Omega \times [0, T] \to \mathbb{R}$. Diffusion coefficients for each species are given by $D_{[\cdot]}$ and chemical reactions are modelled using mass action kinetics [80]. The reaction rate for homogeneous cellular activity ($K_{\mathrm{act}}$) and a spatial reaction rate density, $\mathcal{K}_j$, for the other four reactions. Considering $M$ reaction sites located in $\{\mathbf{x}_i\}_{i=1}^M \in \Omega$, the spatial reaction rates are defined as the product between the classical reaction rates, $K_j$, and Gaussian functions located at those reaction sites with variance $\sigma_i \in \mathbb{R}^+$:

$$\mathcal{K}_j(\mathbf{x}) = \frac{K_j}{\xi}\mathrm{meas}(\Omega)\sum_{i=1}^M \mathcal{G}(\mathbf{x}_i, \sigma_i) \quad j = \{\mathrm{HXK, PYRK, Mito, LDH}\}. \tag{6}$$

$\xi$ is a parameter that ensure the property that $\int_\Omega \mathcal{K}_j \mathrm{d}x = K_j$ and $\mathrm{meas}(\Omega)$ is the area of the domain in 2D or the volume in 3D. The source of GLC is described through a function $J_{\mathrm{in}} : \Omega \times [0, T] \to \mathbb{R}$:

$$J_{\mathrm{in}}(x, t) = \begin{cases} \alpha \in \mathbb{R} & \text{if} \quad (x, t) \in \Omega_{\mathrm{in}} \times [0, T], \quad \text{where} \quad \Omega_{\mathrm{in}} \subset \Omega \\ 0 & \text{otherwise.} \end{cases} \tag{7}$$

Similarly, the degradation of LAC, which is proportional to the amount of LAC in region $\Omega_{\mathrm{out}} \subset \Omega$ is described by function $\eta_{\mathrm{LAC}} : \Omega \times [0, T] \to \mathbb{R}$

$$\eta_{\mathrm{LAC}}(x, t) = \begin{cases} \eta \in \mathbb{R} & \text{if} \quad (x, t) \in \Omega_{\mathrm{out}} \times [0, T], \quad \text{where} \quad \Omega_{\mathrm{out}} \subset \Omega \\ 0 & \text{otherwise.} \end{cases} \tag{8}$$

With this definition, the reaction-diffusion system is given by

$$\begin{cases} \dfrac{\partial[\mathrm{GLC}]}{\partial t} = & D_{[\mathrm{GLC}]}\nabla^2[\mathrm{GLC}] - \mathcal{K}_{\mathrm{HXK}}[\mathrm{GLC}][\mathrm{ATP}]^2 + J_{\mathrm{in}} \\[2mm] \dfrac{\partial[\mathrm{ATP}]}{\partial t} = & D_{[\mathrm{ATP}]}\nabla^2[\mathrm{ATP}] - 2\mathcal{K}_{\mathrm{HXK}}[\mathrm{GLC}][\mathrm{ATP}]^2 + 2\mathcal{K}_{\mathrm{PYRK}}[\mathrm{ADP}]^2[\mathrm{GLY}] \\[2mm] & + 28\mathcal{K}_{\mathrm{Mito}}[\mathrm{PYR}][\mathrm{ADP}]^{28} - K_{\mathrm{act}}[\mathrm{ATP}] \\[2mm] \dfrac{\partial[\mathrm{ADP}]}{\partial t} = & D_{[\mathrm{ADP}]}\nabla^2[\mathrm{ADP}] + 2\mathcal{K}_{\mathrm{HXK}}[\mathrm{GLC}][\mathrm{ATP}]^2 - 2\mathcal{K}_{\mathrm{PYRK}}[\mathrm{ADP}]^2[\mathrm{GLY}] \\[2mm] & + K_{\mathrm{act}}[\mathrm{ATP}] - 28\mathcal{K}_{\mathrm{Mito}}[\mathrm{PYR}][\mathrm{ADP}]^{28} \\[2mm] \dfrac{\partial[\mathrm{GLY}]}{\partial t} = & D_{[\mathrm{GLY}]}\nabla^2[\mathrm{GLY}] + 2\mathcal{K}_{\mathrm{HXK}}[\mathrm{GLC}][\mathrm{ATP}]^2 - \mathcal{K}_{\mathrm{PYRK}}[\mathrm{ADP}]^2[\mathrm{GLY}] \\[2mm] \dfrac{\partial[\mathrm{PYR}]}{\partial t} = & D_{[\mathrm{PYR}]}\nabla^2[\mathrm{PYR}] + \mathcal{K}_{\mathrm{PYRK}}[\mathrm{ADP}]^2[\mathrm{GLY}] - \mathcal{K}_{\mathrm{LDH}}[\mathrm{PYR}] \\[2mm] & - \mathcal{K}_{\mathrm{Mito}}[\mathrm{PYR}][\mathrm{ADP}]^{28} \\[2mm] \dfrac{\partial[\mathrm{LAC}]}{\partial t} = & D_{[\mathrm{LAC}]}\nabla^2[\mathrm{LAC}] + \mathcal{K}_{\mathrm{LDH}}[\mathrm{PYR}] - \eta_{\mathrm{LAC}}[\mathrm{LAC}] \,, \end{cases} \tag{9}$$

where we considered Von Neumann boundary condition to consider no-flux settings at the cell membrane, in agreement with the impermeability of the cell membrane to these intermediates. To characterise the system's behavior, we analysed the equilibrating dynamics towards the steady state from the initial conditions for ATP and ADP concentrations

$$\begin{cases} [\text{ATP}](x, t = 0) \in \mathbb{R} & x \in \Omega \\ [\text{ADP}](x, t = 0) \in \mathbb{R} & x \in \Omega \end{cases}$$

where an initial ATP concentration is required for the initial glycolysis reactions and vanishing concentrations for the other species. To ensure robust simulations, we transformed the RDS into a dimensionless system allowing for convergence over a large parameter range (S8 in S1 Text).

### Immunofluorescence stainings

The PFA-fixed hippocampal samples were cryosectioned into 50–100 $\mu$m thick slices on a sliding freezing microtome (Leica SM2010R). To visualise astrocytes and mitochondria, we co-immunostained the slices against glial fibrillary acidic protein (GFAP) and Tu translation elongation factor mitochondrial (TUFM) respectively. The target-binding primary antibodies used here were Anti-GFAP guinea-pig (Synaptic Systems Cat# 173 004, RRID:AB_10641162) at a dilution of 1:500, and Anti-TUFM mouse (Atlas Antibodies Cat# AMAb90966, RRID: AB_2665738) at a dilution of 1 : 200. The corresponding fluorophore-coupled secondary antibodies used were Alexa Fluor 647-AffiniPure Donkey Anti-Guinea Pig IgG (H+L) (Jackson ImmunoResearch Labs Cat# 706–605-148, RRID:AB_2340476) at a dilution of 1 : 300 and Alexa Fluor 488-AffiniPure Donkey Anti-Mouse IgG (H+L) (Jackson ImmunoResearch Labs Cat# 715–545-150, RRID:AB_2340846) at a dilution of 1 : 400. We followed a previously published protocol [30] with the exception of a double incubation with primary antibodies for the TUFM staining.

### Image acquisitions

High-resolution confocal images with 0.333 $\mu$m z-step were acquired using a Leica DMi8 microscope with a 93X glycerol objective and LAS X software (Leica Microsystems). The region of interest was fixed on the hippocampal subregion CA1.

### Image pre-processing

The surface function of Imaris 9.6.0 software was used to segment GFAP staining to produce astrocyte morphology 3D reconstructions. The surface grain size parameter was set to 0.3 $\mu$m for the segmentation of astrocyte morphology. Upon segmentation of the GFAP signal of the entire image, we manually selected the astrocyte of interest and removed all other non-relevant segmentation structures. The spots function was used to segment TUFM staining. The estimated spots diameter was set to 0.2 $\mu$m. To select only the mitochondria of interest (corresponding to the astrocyte of interest) we applied the filter of the spots function called 'Shortest Distance to Surface' [segmented astrocyte]. In the control astrocyte, some mitochondria of interest were not automatically selected by this filter setting, because they were too far away from the segmented surface, however part of the astrocyte, notably in the cell soma. To include these mitochondria into the analysis, a second filter was applied twice by selecting the central mitochondria of the soma compartment and applying 'Shortest Distance to Surface' function.

The direct use of the astrocytic segmented images as domain for our simulation would require a mesh fine enough to capture the thin branches of the cellular structure. This would

mean billions of quality finite elements, with a good aspect ratio. In literature, this problem was addressed by refining the mesh in critical regions [81]. However, in our case, this would require refining all branches. We overcome these issues by additional image pre-processing where we dilated and down-sampled the binary images. These two steps enlarged the thin branches and avoid discontinuities when we map the images to the finite element mesh. These steps are not critically affecting the real morphology of the astrocytes and might actually address partially the GFAP staining limitation. Moreover, we impose the astrocytic volume in the simulations to be equal to the one of the segmented images obtained with Imaris. Eventually, we obtained the final segmented images (*f*) by labelling the voxels inside (−1), outside (1) and on the boundary (0) of the astrocytes.

Before applying the same steps to the binarised segmented mitochondrial images, we applied a convolution filter to smooth the voxels. To extrapolate the information about mitochondrial density, we selected all connected components in the images and for each of them, we identified the center and the radius of the circle that contains such component.

## Numerical methods

To solve numerically the RDS, the first step is to convert Eq 9 into a corresponding weak form [82]. Then, we discretise the weak form both in time and space. We discretise the time derivative using a finite difference method (backward Euler) [83] and the spatial domain by finite elements [84] and cut finite elements [85]. The 2D experiments were solved using classical finite element methods based on FENICS [86, 87], while the 3D experiments were solved using CUTFEM [33, 85]. Since the weak RDS formulation is non-linear, we linearised it and used a Newton-Raphson algorithm to iteratively solve the problem. The linear system at each time step of the Newton-Raphson algorithm was solved using standard linear solver from the *PETSc* library. For further details and numerical parameters see S9, S10 and S11 in S1 Text, respectively.

## Physiological model parameters

The parameters used in our model are given in Table 1. The diffusion parameters were chosen for ATP and ADP following [38], for GLC based on [37] and for the other species based on the Polson method [39, 40].

The calibration of the reaction rates has been done in accordance with the steady states of the ODE system [24] associated with Eq (9). For $J_{in}$ we used the maximum transport rate of GLC from [24]. For $J_{out}$ we used the maximum transport rate of LAC but divided it for the steady state since we required our transport of LAC to be proportional to the local concentration of LAC inside the cell.

## Supporting information

**S1 Text.** **S1**: Stability analysis. **S2**: Spatial arrangements for 2D simulations in retangular shape. **S3**: Significance test for 2D realisation. **S4**: Local behaviour of ATP, ADP and PYR in maximum and minimum energised reaction site settings. **S5**: Simulating higher cellular activity in the perisynapses and more lactate efflux loci. **S6**: Spatial arrangement for 3D simulations. **S7**: Additional Figure for AD simulations. **S8**: Dimensionless system. **S9**: Detail on numerical methods for 2D simulations. **S10**: Details on numerical methods for 3D simulations. **S11**: Numerical Parameters.
(PDF)

**S1 Movie. GLC control (C) astrocyte.**
(MP4)

**S2 Movie. ATP control (C) astrocyte.**
(MP4)

**S3 Movie. ADP control (C) astrocyte.**
(MP4)

**S4 Movie. GLY control (C) astrocyte.**
(MP4)

**S5 Movie. PYR control (C) astrocyte.**
(MP4)

**S6 Movie. LAC control (C) astrocyte.**
(MP4)

**S7 Movie. GLC reactive (EAD) astrocyte.**
(MP4)

**S8 Movie. ATP reactive (EAD) astrocyte.**
(MP4)

**S9 Movie. ADP reactive (EAD) astrocyte.**
(MP4)

**S10 Movie. GLY reactive (EAD) astrocyte.**
(MP4)

**S11 Movie. PYR reactive (EAD) astrocyte.**
(MP4)

**S12 Movie. LAC reactive (EAD) astrocyte.**
(MP4)

## Acknowledgments

AS and ME would like to acknowledge the important role of the late David J. Galas for his vision and friendship and particularly for his role in seeding the interdisciplinary collaboration, which resulted in this work. The authors would like to thank Corrado Ameli for his help in the segmentation of the control astrocyte and Jack S. Hale for fruitful discussions including the sensitivity analysis.

## Author Contributions

**Conceptualization:** Mark H. Ellisman, Stéphane P. A. Bordas, Alexander Skupin.

**Data curation:** Sonja Fixemer.

**Formal analysis:** Sofia Farina, Valérie Voorsluijs, Alexander Skupin.

**Investigation:** Sofia Farina, Valérie Voorsluijs, Alexander Skupin.

**Methodology:** Sofia Farina, Valérie Voorsluijs, Susanne Claus, Stéphane P. A. Bordas, Alexander Skupin.

**Software:** Sofia Farina.

**Supervision:** Valérie Voorsluijs, David S. Bouvier, Stéphane P. A. Bordas, Alexander Skupin.

**Visualization:** Sofia Farina.

**Writing – original draft:** Sofia Farina, Valérie Voorsluijs, Alexander Skupin.

**Writing – review & editing:** Sofia Farina, Valérie Voorsluijs, Sonja Fixemer, David S. Bouvier, Susanne Claus, Mark H. Ellisman, Stéphane P. A. Bordas, Alexander Skupin.

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
