## [Decision Letter · Decision Letter 0]

12 May 2023

Dear Skupin,

Thank you very much for submitting your manuscript "Mechanistic multiscale modelling of energy metabolism in human astrocytes indicates morphological effects in Alzheimer’s Disease" for consideration at PLOS Computational Biology.

As with all papers reviewed by the journal, your manuscript was reviewed by members of the editorial board and by several independent reviewers. In light of the reviews (below this email), we would like to invite the resubmission of a significantly-revised version that takes into account the reviewers' comments.

We cannot make any decision about publication until we have seen the revised manuscript and your response to the reviewers' comments. Your revised manuscript is also likely to be sent to reviewers for further evaluation.

Sincerely,

Hugues Berry

Academic Editor

PLOS Computational Biology

Thomas Serre

Section Editor

PLOS Computational Biology

Reviewer's Responses to Questions

**Comments to the Authors:**

Reviewer #1: The manuscript by Sofia Farina and colleagues describes their modeling research focusing on astroglial energy metabolism in the context of neurodegenerative diseases, when astrocytes are hypothesized to loose their ability to sustain neurons with lactate.

The authors describe a minimalistic lumped model of cell metabolism and look at steady-state average metabolite concentrations depending on spatial arrangement of the four modeled enzymatic complexes or mechanisms, namely hexokinase (HXK), pyruvatkinase (PRK), lactate dehydrogenase (LDH), and oxydative phosphorilation in mitochondria (Mito). They start with arguably over-simplistic spatial profiles and proceed to realistic astrocyte morphologies extracted from experimental data.

While the approach is indeed original and provides interesting insights, one has to comment on several issues with the manuscript in its current form.

** General remarks and model organization

– The assumption that cellular activity should occur homogeneously inside the domains (p. 9, line 165) is not sufficiently grounded; one can expect e.g. that perisynaptic processes can spend more energy restoring ionic gradients to fuel neurotransmitter uptake.

– Narrow spatial confinements of GLC influx and LAC outflux have to be justified. While GLC influx can be speculated to be confined to astro. endfoot, LAC outflux seems more likely to be sprinkled over multiple perisynaptic loci.

– what dictates the choice of log-normal distribution of enzymatic complexes or mitochondria?

– while at first a competition for pyruvate between Mito and LDH is mentioned (p. 11, line 199), it is somehow transformed into Mito-PYRK competition in p. 14, line 251, going as far as a suggestion that PYRK “inhibits mitochondrial activity”. This is counter-intuitive, because PYRK is the source of substrate for Mito.

– It is further unclear why the lowest levels of ATP are observed in the arrangements with all mitochondria grouped near HXK and PYRK. Shouldn’t a placement of Mito closer to the site of substrate production enhance ATP production? An alternative explanation of the observed results could be in the line that if the mitochondria are grouped, then any runaway PYR molecule will have a chance to be converted to LAC at the far end of the elongated cell, while spreading out mitochondria along the cell axis will reduce the change for PYR to escape its oxyphospho fate.

– The fact that high ATP levels inhibits glycolysis, discussed in the ATP:ADP-related papers (e.g. Maldonado, Lemaster, Mitochondrion 2014) in the context of Warburg pathway is not accounted for in the model, but it can be rather important for the outcome.

– The justification for the spatial arrangements in Figs. 3-4, especially the artificial second cluster of mitochondria near the LDH-enriched site in the “polarized” arrangement should be spelled out more explicitly.

– The observation (and explanation of) that non-reactive astrocyte exports more LAC than the more ramified reactive one is counterintuitive. This looks like noting that a less ramified pipeline system will deliver more water to a town than a more ramified one due to longer pathway of water to each household. It is easy to imagine that the more ramified astrocyte will have more LAC outlets to tap lactate to the tissue and will export more of it, so it’s a question of the number of export regions.

– Concerning the spatial arrangement of LAC export sites. First, just four sites of LAC export per astrocyte seems like a severe underestimate. Second, I would suggest to address IHC images of LAC transporters in astroglial membrane (e.g. how uniform vs clustered they are).

– Overall, I would suggest to shorten the artificial template part of the research (e.g. by dropping the circle and star arrangements entirely) to emphasize the really relevant part with experiment-based morphologies. For example, by comparing more than one cells from each group. This would also allow for a more clear description of and justification for the metabolic challenges simulated in the AD case. Additionally, the boundary between the simulation and physiological predictions for real cells seems just a bit too thin in the text allowing for statements like “The only cell that reaches a critical unhealthy state is the EAD condition”, “When cells lack GLC, they become more egoistic and produce more ATP”, “single AD characteristic does not lead to an unhealthy cell” etc. I would advocate for a more clear distinction between the (arguably, simplistic) simulation and the physiological implications from the simulation results.

** Figures and results representation

– Fig. 1:

- [p. B,C]: what is the physical meaning and what are the units for the colorbar?

- [p. B,C]: why are some of the x-axis and y-axis values negative (the template also being not centered)?

- [Caption]: may be it’s better to denote different spatial organizations as Arrangement 1 and Control instead of Location 1 and control?

- [general]: what justifies combination of a pair of mechanisms in a single vertex? i.e. why not make 4 vertices for each of the complexes?

– Fig. 3: to increase clarity, it’s would probably help to plot the distributions with empty background, in a “stair” style or alpha channel. Otherwise some parts of the distributions are occluded.

– Fig. 4.

- It is unclear, what exactly is plotted in panel C and how it’s related to spatial arrangements in panel D.

- Caption: “mitochondria activation” should be “mitochondria activity”

– Fig. 5. The reconstruction of astrocyte morphologies seems imperfect, for example a large chunk of the endfoot from protoplasmic astrocyte appears to be missing (and it’s unclear where’s the endfoot in the AD astrocyte at all).

– Fig. 7. [panel A]: connecting lines have no meaning, please consider to remove them

** References to literature

- There is no year or year is poorly formated: Refs. 7, 11, 22, 31

- Ref 19 (Almeida et al PNAS 2001) is cited in support of the notion that ATP:ADP ratio in healthy cells should be > 1. However, this point is not made in the publication.

Reviewer #2: This manuscript presents a mechanistic model of energy metabolism in spatially resolved implementation considering idealised as well as realistic morphology of the cell (astrocyte). It is well written and should be of interest to the audience of the journal. As the modelling is complex and the numerical implementation involving a number of assumptions I do believe that the clarity could be improved by providing more details on the following:

1) There are various sources of uncertainty - such as code uncertainty, parameter uncertainty, model discrepancy, and observation error - inherent to any computational model analysis. Prior to undertaking model-based inference it is important to consider such uncertainties and quantify as many as possible. I would recommend that the authors at least comment on this and ideally provide some uncertainty estimates for example, parameter sensitivity measures.

2) The choice for modelling metabolism by considering a rather coarse representation of metabolic reactions could be further justified.

3) The choices of uniform, normal and log-normal distributions for the reaction sites also need to be justified in more detail. Why these particular distributions? How realistic is this?

4) I am not convinced in the following statement (page 13, lines 219-222): "Interestingly, the Polarised log N (2) configuration exhibits a very wide range for both concentrations with significantly different average values also in comparison with the Polarised configuration indicating the importance of mitochondrial distribution." How does this indicate "the importance of mitochondria distribution" and why? Could you please elaborate.

5) The choice of no-flux boundary conditions used when simulating the reaction-diffusion system needs to be justified. How biologically realistic is this assumption? Why is it acceptable?

6) Typo on page 30, line 554 - "we discretise..."  "We discretise..."

Reviewer #3: First and foremost, let me congratulate the authors on a very well performed and presented study. The introduction is nicely written, clear and convincing. I especially appreciate the authors statement: "The underlying assumption that diffusion and reaction rates of metabolism are large enough to smear out spatial aspects are challenged by the complex morphology of astrocytes and an increasing amount of evidence for relocation of enzymes and other reaction sites in different conditions". The numerical experiments are well performed and results clearly described.

Major comments

I strongly suggest that

* The authors revisit the title. The current title, especially the "indicates morphological effects" is unnecessary unclear/weak.

* The authors revisit the last sentence of the Author summary. As in the case of the title, I find the phrase "fundamental to ensure robustness by buffering effect" unclear.

* The authors considers whether Table 2 should be included as is, and whether the numbers are accurate. As is, the signal-to-noise ratio seems low. The table could be moved to the Supplementary, or the non-zero values described in the text.

* The authors include an estimate/numerical experiment relating to the accuracy of the numerical approximations for image-based geometries in Results. In the Supplementary, the authors state that "Convergence study were done extensively in the 2 dimensional experiments." I would encourage the authors to provide more detail. Furthermore, the question of how numerically accurate/converged/resolved the reported results from the image-based geometries are remains. I would encourage the authors to address this vital question.

* The authors revise the Discussion. The authors summarize their own results at length, but only marginally compare/discuss their findings with previous observations from the (modelling, experimental and/or clinical) literature. This discussion should be extended and be made more specific.

Minor comments:

The authors should consider the following minor points

- L9: While [it] is known ...

- L11: Replace "insults" by e.g. trauma

- L36: addressed in [the] literature

- P16 and Fig 5: Inconsistent labelling of subfigures (a-c vs A-C).

**Have the authors made all data and (if applicable) computational code underlying the findings in their manuscript fully available?**

Reviewer #1: **No: **The provided link to the code and data (https://bitbucket.org/sofiafarina/metabolic_model_astrocytes/src/master/) results in 404 error

Reviewer #2: **No: **I get the following error message following the link provided in the Data Availability section "Repository not found. You may not have access to this repository or it no longer exists in this workspace. If you think this repository exists and you have access, make sure you are authenticated."

Reviewer #3: Yes

PLOS authors have the option to publish the peer review history of their article (what does this mean?). If published, this will include your full peer review and any attached files.

Reviewer #1: No

Reviewer #2: No

Reviewer #3: No
---

## [Decision Letter · Decision Letter 1]

25 Aug 2023

Dear Skupin,

We are pleased to inform you that your manuscript 'Mechanistic multiscale modelling of energy metabolism in human astrocytes reveals the impact of morphology changes in Alzheimer's Disease' has been provisionally accepted for publication in PLOS Computational Biology.

Best regards,

Hugues Berry

Academic Editor

PLOS Computational Biology

Thomas Serre

Section Editor

PLOS Computational Biology

Reviewer's Responses to Questions

**Comments to the Authors:**

Reviewer #1: The authors have thoroughly addressed all points raised in the first review round. I agree with their reasoning. In my view, the manuscript can be accepted for publication in the current state.

Reviewer #2: The authors have addressed my comments satisfactory.

Reviewer #3: All comments have been address satisfactorily.

**Have the authors made all data and (if applicable) computational code underlying the findings in their manuscript fully available?**

Reviewer #1: Yes

Reviewer #2: Yes

Reviewer #3: Yes

PLOS authors have the option to publish the peer review history of their article (what does this mean?). If published, this will include your full peer review and any attached files.

Reviewer #1: No

Reviewer #2: **Yes: **Krasimira Tsaneva-Atanasova

Reviewer #3: No

---

## [Editor Report · Acceptance letter]

15 Sep 2023

PCOMPBIOL-D-23-00412R1 

Mechanistic multiscale modelling of energy metabolism in human astrocytes reveals the impact of morphology changes in Alzheimer's Disease

Dear Dr Skupin,

I am pleased to inform you that your manuscript has been formally accepted for publication in PLOS Computational Biology. Your manuscript is now with our production department and you will be notified of the publication date in due course.

With kind regards,

Timea Kemeri-Szekernyes
